# Adversarial attacks and adversarial robustness in computational pathology

Narmin Ghaffari Laleh[1], Daniel Truhn[2], Gregory Patrick Veldhuizen[3], Tianyu Han [4], Marko van Treeck[1], Roman D. Buelow [5], Rupert Langer[6,7], Bastian Dislich[6], Peter Boor [5], Volkmar Schulz [4,8,9,10] & Jakob Nikolas Kather [1,3,11,12,13] ✉

Artificial Intelligence (AI) can support diagnostic workflows in oncology by aiding diagnosis and providing biomarkers directly from routine pathology slides. However, AI applications are vulnerable to adversarial attacks. Hence, it is essential to quantify and mitigate this risk before widespread clinical use. Here, we show that convolutional neural networks (CNNs) are highly susceptible to white- and black-box adversarial attacks in clinically relevant weakly-supervised classification tasks. Adversarially robust training and dual batch normalization (DBN) are possible mitigation strategies but require precise knowledge of the type of attack used in the inference. We demonstrate that vision transformers (ViTs) perform equally well compared to CNNs at baseline, but are orders of magnitude more robust to white- and black-box attacks. At a mechanistic level, we show that this is associated with a more robust latent representation of clinically relevant categories in ViTs compared to CNNs. Our results are in line with previous theoretical studies and provide empirical evidence that ViTs are robust learners in computational pathology. This implies that large-scale rollout of AI models in computational pathology should rely on ViTs rather than CNN-based classifiers to provide inherent protection against perturbation of the input data, especially adversarial attacks.

Artificial intelligence (AI) with deep neural networks can extract clinically relevant information from digitized pathological slides of cancer[1–3]. Over the last several years, hundreds of studies have shown that diagnostic, prognostic, and predictive models can achieve accuracy which is comparable with gold standard methods[4–7]. Most studies investigate applications in cancer diagnostics and treatment, where a pathological diagnosis is a cornerstone and slides are ubiquitous[8–10]. It is widely expected that AI systems will increasingly be used in clinical practice for cancer diagnostics and biomarker identification over the coming years[11,12]. Ultimately, such AI systems have the potential not only to make existing workflows more efficient, but also enable physicians to recommend improved treatment strategies for cancer patients[13–16].

Considering this, it is crucial to ensure that the AI systems are robust before they are used in diagnostic routines. AI systems should be resilient to subtle changes in input data and yield a stable performance, even when the input signal is noisy. In particular, this includes adversarial attacks to the input signal, i.e., willful modifications to the input data by a malicious actor. Adversarial attacks are a vulnerability of AI systems which is a concern in many domains[17]. The most common of these attack types are called white-box attacks. In such attacks, the adversary has full access to the model's parameters[18]. In contrast, black-box attacks hide the original model from the attacker. Adversarial changes to the original data are usually undetectable to the human eye but are disruptive enough to cause AI models to misclassify samples.

Cybersecurity is highly relevant for the development and regulation of software in healthcare[19]. AI systems in healthcare are particularly vulnerable to adversarial attacks[20]. This poses a significant security risk: predictions of AI systems in healthcare have potentially major clinical implications, and misclassifications in clinical decision-support systems could have lethal consequences for patients. Thus, AI systems in healthcare should be particularly robust against any attacks. Yet, in computational pathology, only very few studies have explored adversarial attacks[21]. To date, no established strategy has been developed to make AI systems in the field of digital pathology robust against such attacks. The development of attack-resistant AI systems in pathology is, therefore, an urgent clinical need, which should ideally be resolved before these systems are widely deployed in diagnostic routine.

To date, convolutional neural networks (CNNs) are by far the most used type of deep neural network in digital pathology[22,23]. CNNs are capable of capturing high-level features such as edges from input data by applying various kernels throughout the training process. As of late 2020, vision transformers (ViTs) have emerged as an alternative to CNNs. ViTs use lower-dimensional linear embeddings of the flattened small patches extracted from the original image as input to a transformer encoder[24]. Unlike CNNs, ViTs are not biased toward translation-invariance and locally restricted receptive fields[25]. Instead, their attention mechanism allows them to learn distal as well as local relationships. Although ViTs have outperformed CNNs in some non-medical prediction tasks, the uptake of this technology is slow in medical imaging. To date, only very few studies have investigated the use of ViTs in computational pathology[23,26,27]. Technical studies have described improved robustness of ViTs to adversarial changes to the input data, but this has not been explored in medical applications[28–32].

In this study, we investigated the robustness of CNNs in computational pathology toward different attacks and compared these results to the robustness of ViTs. Additionally, we trained robust neural network models and evaluated their performances against the white- and black-box attacks. We analyzed the attack structure for both models and investigated the reasons behind their performances. We validated our results in two clinically relevant classification tasks in independent patient cohorts[33–36]. This study adheres to the MI-CLAIM50 checklist (Suppl. Table 1).

## Results

### CNN and ViT perform equally well on clinically relevant classification tasks

Prediction of the main histological subtypes of renal cell carcinoma (RCC) into clear cell carcinoma (ccRCC), chromophobe carcinoma (chRCC), and papillary carcinoma (papRCC) is a widely studied task in computational pathology[23,33]. We trained ResNet, a convolutional neural network (CNN, Fig. 1A) and a ViT (Fig. 1B) on this task on TCGA-RCC ($N = 897$ patients, Suppl. Fig. 1A). The resulting classifiers performed well on the external test set AACHEN-RCC ($N = 249$, Suppl. Fig. 1B), reaching a mean area under the receiver operating curve (AUROC) of 0.960 [±0.009]. ViT reached a comparable AUROC of 0.958 [±0.010] (Fig. 1C and Suppl. Table 2), which was on par with and not significantly different from the ResNet ($p = 0.98$). The image tiles which were assigned the highest scores showed typical patterns for each histological subtype, demonstrating that ResNet and ViT can learn relevant patterns and generalize to an external validation cohort (Fig. 1D). In addition, we evaluated the baseline performance of CNN and ViT on subtyping of gastric cancer[37,38]. When trained on the TCGA-GASTRIC cohort ($N = 191$ patients, Suppl. Fig. 1C) and tested on the BERN cohort ($N = 249$, Suppl. Fig. 1D), CNN and ViT achieved mean AUROCs of 0.782 [±0.014] and 0.768 [±0.015] respectively (Fig. 1E and Suppl. Table 2). Again, the highest-scoring tiles showed morphological patterns which are representative of the diffuse and intestinal subtype (Fig. 1F)[39,40]. Together, these data are in line with the

previous evidence[23] and show that CNNs and ViTs perform equally well for weakly-supervised classification tasks in our experimental pipeline.

### CNNs are susceptible to multiple adversarial attacks

We attacked CNNs with adversarial attacks (Fig. 2A), evaluating white-box and black-box attacks (Fig. 2B). By default, we used the most commonly used gradient-based attack, Projected Gradient Descent (PGD), and additionally tested five other types of adversarial attacks (Fast Gradient Sign Method [FGSM], Fast Adaptive boundary [FAB], Square attacks, AutoAttack [AA], and AdvDrop, Fig. 2C). We found that with an increasing attack strength ε, the amount of visible noise on the images increased (Fig. 2D). We quantified this in a blinded observer study and found that the detection threshold for adversarial attacks was ε = 0.19 for ResNet models and ε = 0.13 for ViT (Suppl. Table 3 and Suppl. Fig. 2A, B). With increasing attack strength, the classifier performance of a ResNet CNN on the test set decreased. Specifically, we attacked with PGD with a low (ε = 0.25e-3), medium (ε = 0.75e-3), and high (ε = 1.50e-3) attack strength. The AUROC for RCC subtyping by ResNet dropped from a baseline of 0.960 to 0.919, 0.749, and 0.429 (Fig. 3A and Suppl. Table 4). For the secondary classification task, subtyping gastric cancer, the CNN models were even more susceptible to adversarial attacks. Here, the PGD completely degraded classification performance. The AUROC reached by the CNN dropped from a baseline of 0.782 to 0.380, 0.029, and 0.000 for the images attacked with low, medium, and high ε (Fig. 3B and Suppl. Table 5). Together, these data show that CNNs are highly susceptible to adversarial attacks in computational pathology.

### Adversarially robust training partially hardens CNNs

We subsequently investigated two possible mitigation strategies to rescue CNN performance. First, we evaluated adversarially robust training, in which PGD is applied to the training dataset so that CNN can learn to ignore the noise patterns. Although training a CNN with PGD-attacked images (ε = 1.50e-3) slightly reduced the RCC classification performance from baseline from 0.960 to 0.954 (Suppl. Table 2), it improved the model's robustness to attacks. For the PGD attack at inference, this adversarially robustly trained CNN yielded an average AUROC of 0.951, 0.944, and 0.932 for low, medium, and high ε, respectively (Fig. 3A and Suppl. Table 6). Second, we investigated if the effect of adversarially robust training of CNNs could be enhanced by using a dedicated technique, dual-batch-normalized (DBN). The baseline performance of this model was an AUROC of 0.946 [±0.028] ($p = 0.58$) for RCC classification, which was not significantly inferior to the original model (Suppl. Table 2). When we attacked the test dataset with the PGD attack, DBN-CNN conveyed good protection at inference, but did not beat the normal adversarially robust training (Fig. 3A and Suppl. Table 6). In the secondary prediction task, adversarially robust training slightly lowered the classification accuracy at baseline (on non-attacked images) from 0.782 [±0.014] to 0.754 [±0.012], but mitigated the vulnerability to attack, resulting in AUROCs of 0.731, 0.679, and 0.595 for low, medium and high ε (Suppl. Table 7). Together, these data show that the attackability of CNNs can be partly mitigated by adversarially robust training. Dual batch normalization (DBN) did not convey any additional robustness to CNNs.

### ViTs are inherently robust to adversarial attacks

Next, we attacked ViTs with adversarial attacks. We found that they were relatively robust against adversarial attacks without any adversarial pretraining and without any modifications to the architecture. For low, medium, and high PGD attack strengths in RCC classification, ViT AUROCs were slightly reduced from a baseline of 0.958 to 0.944, 0.908, and 0.827 (Suppl. Table 4), but ViT was significantly more robust than Resnet ($p = 0.06$, 0.04, and 0.01). For the secondary prediction task of gastric cancer subtyping, the baseline performance was lower for all classifiers when compared to RCC (Fig. 3B). Also in this

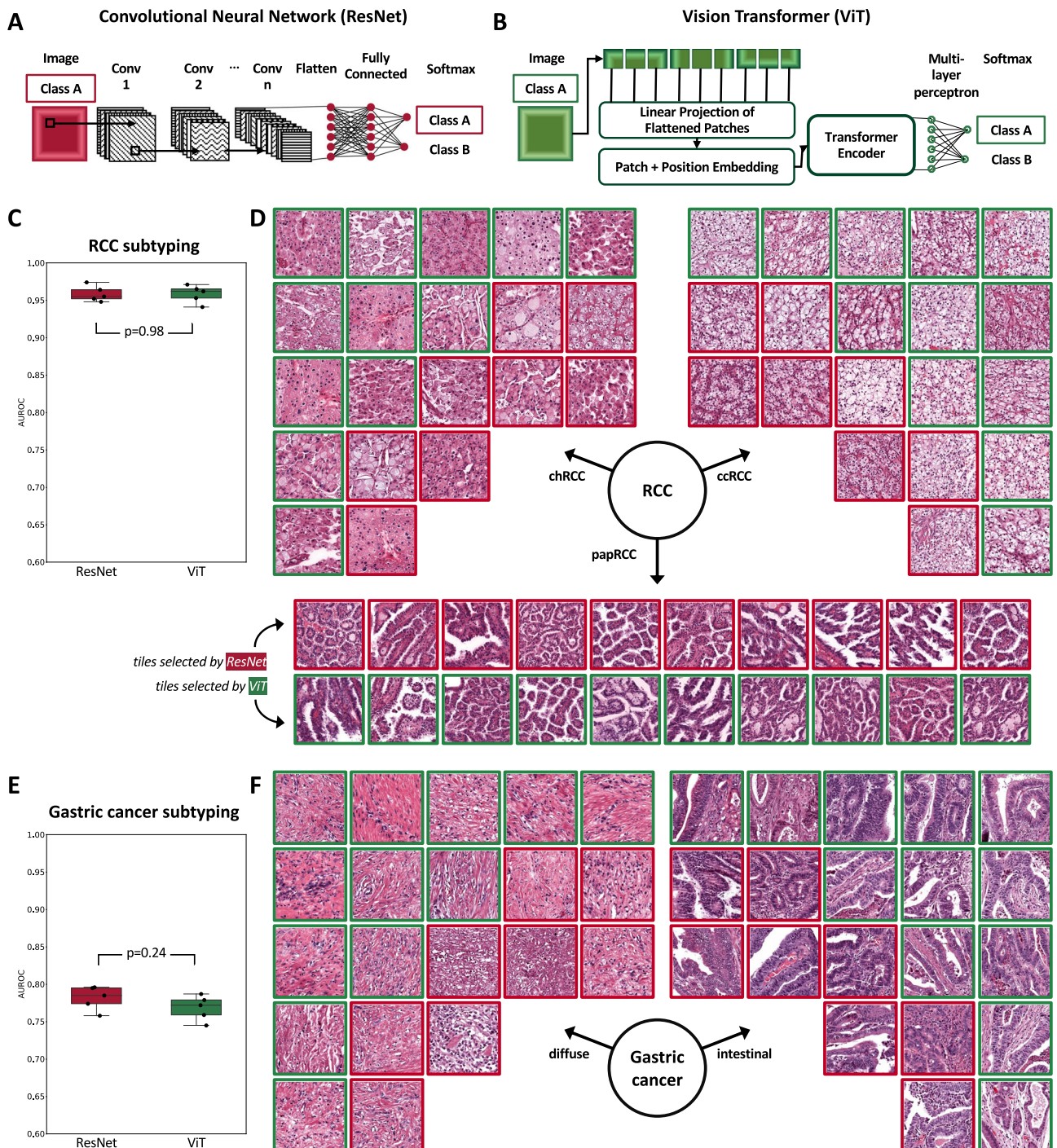

**Fig. 1 | Cancer subtyping with Deep Learning. A** Image classification with ResNet, **B** with a Vision Transformer (ViT). **C** Area under the receiver operating curve (AUROC) for subtyping of renal cell carcinoma (RCC) into clear cell (cc), chromophobe (ch), and papillary (pap). The box shows the median and quartiles of five repetitions (points) and the whiskers expand to the rest of the distribution ($n = 249$ patients). We used a two-sided $t$-test without adjustments for the performance comparison between the two models. **D** Representative highly scoring image tiles for RCC, as selected by ResNet and ViT. **E** AUROC for subtyping gastric cancer into diffuse and intestinal. The box shows the median and quartiles of five repetitions (points) and the whiskers expand to the rest of the distribution ($n = 249$ patients). We used a two-sided $t$-test without adjustments for the performance comparison between the two models. **F** Highly scoring image tiles for gastric cancer, as selected by ResNet and ViT.

task, ViTs were significantly more robust to attacks than ResNet ($p < = 0.01$ for low, medium and high attack strength, Suppl. Table 5). Training a ViT in an adversarially robust way slightly reduced the baseline performance for RCC classification from 0.958 [±0.01] to 0.938 [±0.007] (Fig. 3A), and reduced the performance of ViT under a low-intensity PGD attack from 0.944 [±0.011] to 0.932 [±0.007]. However, for medium and high-intensity attacks, adversarially robust

training was beneficial for ViTs, slightly increasing the AUROC from 0.908 [±0.015] to 0.922 [±0.01] and from 0.827 [±0.032] to 0.906 [±0.016], respectively (Suppl. Tables 4, 6). Similarly, in the gastric cancer classification task, adversarially robust training hardened ViTs: they only slightly reduced their baseline AUROC of 0.737 to 0.724, 0.699, and 0.657 under low, medium, and high-intensity attacks, respectively (Suppl. Table 7). Next, we investigated whether the

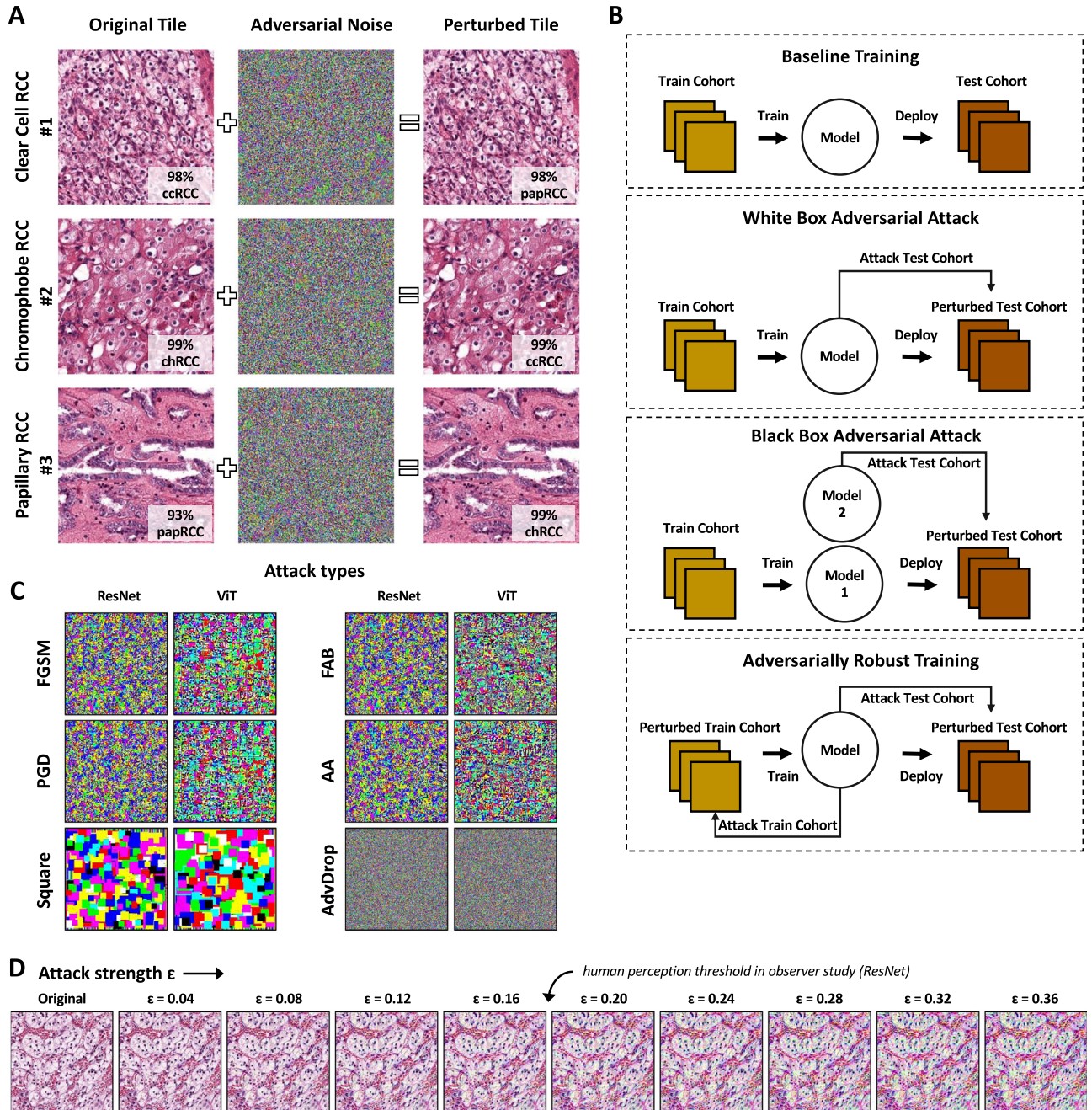

**Fig. 2 | Adversarial attacks on computational pathology. A** Adversarial attacks add noise to the image and flip the classification of renal cell carcinoma (RCC) subtyping into a clear cell (cc), chromophobe (ch), and papillary (pap). The model's prediction confidence is shown on each image. **B** Experimental design for the baseline (normal) training, white-box, and black-box attacks and for adversarially robust training. **C** Different attack algorithms yield different noise patterns. We used the Fast Gradient Sign Method (FGSM), Projected Gradient Descent (PGD), Fast Adaptive boundary (FAB), Square attacks, AutoAttack (AA), and AdvDrop. **D** The attack strength ε increases the amount of noise which is added to the image. The average threshold for human perception is ε = 0.19 for ResNet.

improved higher robustness of ViTs compared to CNNs extended to other types of white and black-box attacks. To this end, we selected 450 tiles from the RCC subtyping task and calculated the attack success rate (ASR) for an overall 6 attacks under low, medium, and high attack strength (ε = 0.25e-3, 0.75e-3, and 1.50e-3) (Table 1). For all six types of attacks, in baseline models and adversarially trained models, ViTs had a lower (better) ASR in the majority of experiments. For baseline models, ViT outperformed ResNet for all the attack types and for all predefined attack strengths ε (Suppl. Fig. 3). For adversarially trained models, the margin was smaller, but ViT still outperformed ResNet in 9 out of 24 experiments (Table 1). In addition, we investigated whether the higher robustness of ViT compared to ResNet was due to its pretraining

on a larger image set or its higher number of parameters. To this end, we repeated our experiments with another CNN model, the BiT, which is similar to the original ResNet, but has more parameters and is trained on more data during pretraining. We found that BiT was even more susceptible to adversarial attacks than the baseline ResNet (Table 1) and was similarly inferior to ViT for sub-visual attack strengths ε. Finally, we evaluated attacks with a very high ε value of 0.1 (Table 1), which resulted in a severe performance reduction for all models. However, because 0.1 is at the threshold for human perception, these attacks are potentially of low practical relevance. In contrast, attacks in the low sub-visual range (e.g., ε 1.5e-3, as used by us and by the previous studies[41]) are very hard to detect and still detrimental to the

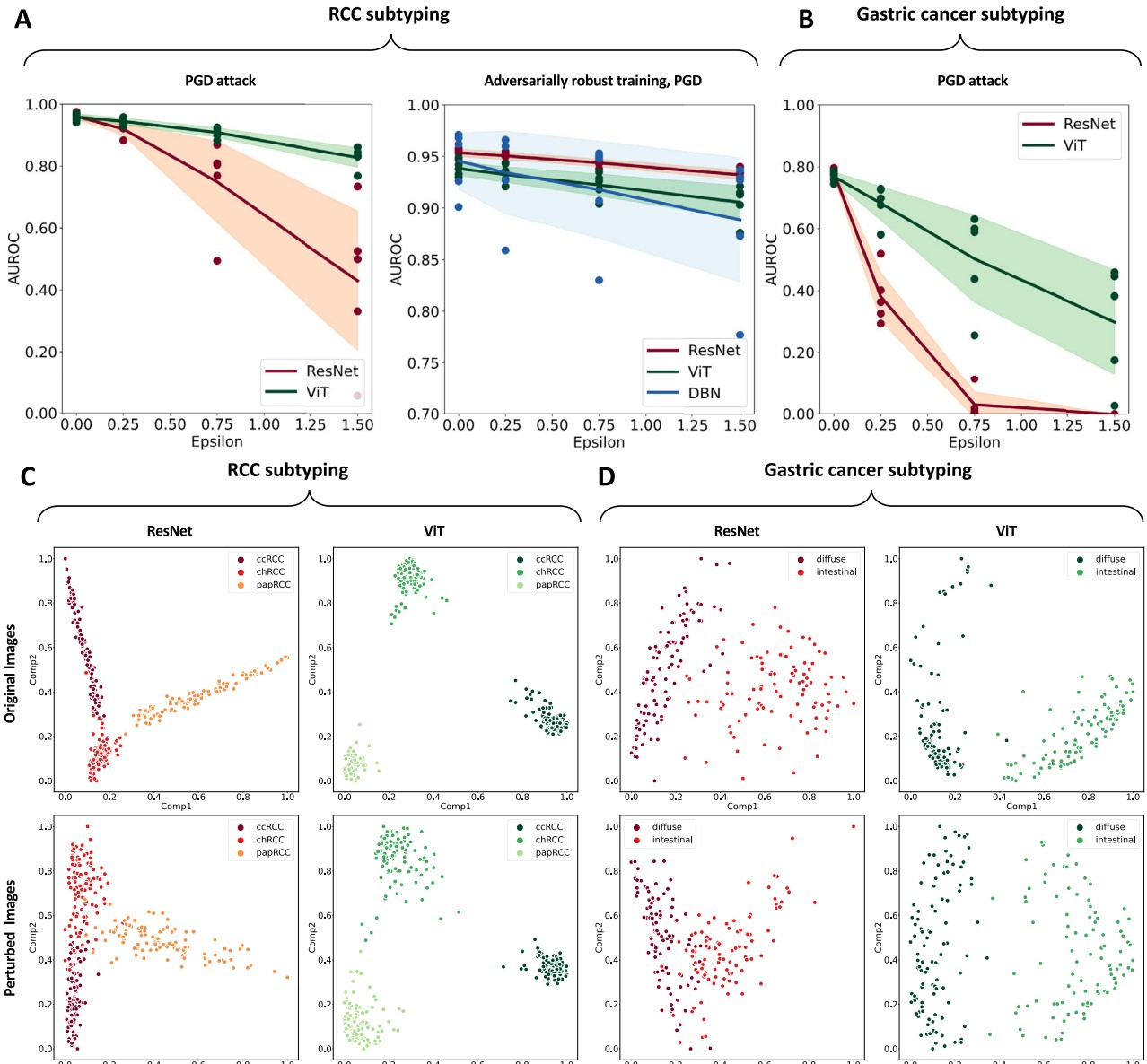

**Fig. 3 | Vision transformers are more robust to adversarial attacks than convolutional neural networks. A** Micro-averaged AUROC for ResNet and ViT under PGD attack for RCC subtyping without (left) and with (right) adversarially robust training. Epsilon * 10E-3. This figure shows the mean AUROC of five experiments ± the standard deviation. **B** AUROC for ResNet and ViT for gastric cancer subtyping. ε * 10e-3. This figure shows the mean AUROC of five experiments ± the standard deviation. **C** First two principal components of the latent space of ResNet and ViT before (original) and after the attack (perturbed) for RCC subtyping, for 150 highest-scoring image tiles. ViT has better separation of the clusters before the attack and its latent space retains its structure better after the attack. **D** Latent space for the gastric cancer subtyping experiment.

performance of convolutional neural networks, placing these attacks in the focus of adversarially robust model development.

**Mechanism of ViT robustness against adversarial attacks**

To identify potential reasons for this higher robustness of ViTs towards adversarial attacks, we analyzed the adversarial noise obtained with white-box attacks on ViTs and ResNets. Quantitatively, we found that the magnitude of the gradients was consistently lower for ViT than for ResNet (Suppl. Fig. 4A). Qualitatively, in ViT, we observed a clear patch partition boundary alignment while ResNet patterns were more spatially incoherent (Suppl. Fig. 4B). We conclude that this observation reflects the patch-based nature of ViTs, which causes learned features to contain less low-level information such as lines and edges from an input image and therefore making them less sensitive to high-frequency perturbations. In addition, we analyzed the structure of the latent space of the deep layer activations in ResNet and ViT, after dimensionality reduction with principal component analysis (PCA). We found that for the original images in the RCC classification tasks, the instances in the classes were visually more clearly separated for ViT than for the CNN (Fig. 3C). This was confirmed in the more difficult task of gastric cancer subtyping, in which also a clearer separation was seen (Fig. 3D). Quantitatively, the instances within a given class were aggregated more tightly in the ViT latent space, and the distance between the centers of the classes were larger (Suppl. Table 8). When we attacked the images and used the baseline model to extract the features, the differences were even more pronounced: the ResNet latent space was more de-clustered than the ViT latent space (Fig. 3C, D). Finally, we investigated which regions in input images were assigned high importance by the ResNet and the ViT, respectively, visualizing important regions with Grad-CAM. At baseline, the ResNet

**Table 1 | ViTs are more robust to adversarial attacks than ResNets, as measured by the attack success rate (ASR) for the RCC classification task**

| ε | FGSM | | | PGD | | | Square | | | FAB | | | AutoAttack | | | ε | AdvDrop | | |
|---|---|---|---|---|---|---|---|---|---|---|---|---|---|---|---|---|---|---|---|
| | ResNet | BiT | ViT | ResNet | BiT | ViT | ResNet | BiT | ViT | ResNet | BiT | ViT | ResNet | BiT | ViT | | ResNet | BiT | ViT |
| **Normal models** | | | | | | | | | | | | | | | | | | | |
| 0.25e-3 | 13.33% | 16.44% | **2.22%** | 14.44% | 16.22% | **2.22%** | 5.78% | 2.22% | **0.6%** | 12.67% | 19.78% | **2.00%** | 13.56% | 19.78% | **2.00%** | 20 | 68.67% | 63.11% | **61.56%** |
| 0.75e-3 | 32.67% | 35.56% | **6.44%** | 34.67% | 33.78% | **7.33%** | 13.56% | 7.56% | **2.00%** | 29.78% | 43.56% | **6.00%** | 33.11% | 44.44% | **6.44%** | 40 | 67.56% | 68.44% | **45.11%** |
| 1.50e-3 | 46.00% | 46.00% | **12.89%** | 50.22% | 45.56% | **14.44%** | 24.00% | 15.78% | **3.11%** | 44.44% | 56.44% | **12.00%** | 48.67% | 56.89% | **13.33%** | 60 | 55.78% | 70.00% | **45.11%** |
| 0.1 | 64.22% | 62.00% | 55.11% | 64.00% | 63.33% | 60.89% | 54.89% | 58.00% | 55.78% | 52.00% | 58.00% | 55.11% | 54.89% | 58.00% | 55.78% | - | - | - | - |
| **Adversarially trained models** | | | | | | | | | | | | | | | | | | | |
| 0.25e-3 | **0.70%** | 7.11% | 0.90% | **0.70%** | 7.11% | 0.90% | **0.22%** | 1.33% | 0.44% | **0.70%** | 9.11% | 0.90% | **0.70%** | 9.11% | 0.90% | 20 | 68.89% | 41.78% | 58.22% |
| 0.75e-3 | 2.89% | 16.00% | **2.00%** | 2.89% | 15.33% | **2.00%** | **0.67%** | 2.89% | 0.90% | 2.89% | 23.33% | **2.00%** | 2.89% | 24.44% | **2.00%** | 40 | 75.78% | 50.22% | 63.78% |
| 1.50e-3 | 6.44% | 23.33% | **3.56%** | 6.67% | 20.44% | **3.78%** | 2.00% | 7.56% | **0.90%** | 6.67% | 39.33% | **3.78%** | 6.89% | 41.56% | **3.78%** | 60 | 75.78% | 51.56% | 64.44% |
| 0.1 | 62.00% | **42.67%** | 51.33% | 72.44% | **55.11%** | 60.67% | 61.56% | **47.56%** | 50.89% | 60.89% | **47.55%** | 54.00% | 62.00% | **47.56%** | 54.22% | - | - | - | - |
| Winner | ViT | | | ViT | | | ViT | | | ViT | | | ViT | | | | | | |
| t [sec] | 0.08 s | 0.13 s | 0.19 s | 2.51 s | 3.78 s | 4.36 s | 31.56 s | 47.72 s | 30.16 s | 4.10 s | 4.47 s | 5.09 s | 5.30 s | 3.56 s | 6.74 s | | 5.10 s | 2.14 s | 3.46 s |

The computation time t is the time needed to apply the attack to each image. For pairwise comparisons between ResNet, BiT, and ViT for the same experimental condition, the one with the lower (better) ASR is printed in bold. In this experiment, 450 randomly selected tiles from AACHEN-RCC were used (same tiles for all experiments). The best value in each category is typeset in bold font.

tended to focus on a single region of the input image, while ViT assigned higher importance to multiple image regions. After adversarial attacks, the ResNet region's importance was defocused and included much larger, potentially irrelevant image regions. This effect increased with increasing attack strength ε. In contrast, the important image regions as highlighted by Grad-CAM in a ViT did not visibly change during an attack (Suppl. Fig. 5). Based on these observations, we conclude that the high robustness of ViT towards white-box adversarial attacks, when compared with CNN, is associated with a better separation of distinct classes in the latent space, and a more stable focus on relevant image regions within image tiles.

## Discussion

Machine learning (ML) based software as medical devices (SaMD) can be a target of cyberattacks, which have the potential to cause significant harm[19]. Adversarial attacks can manipulate AI systems into giving false predictions[20]. The number of AI systems used in healthcare is massively increasing[42]. A particularly relevant domain of application is computational pathology, where AI systems have been shown to solve clinically relevant questions in the last few years[4]. Based on these academic developments, advanced AI algorithms have already entered the market. Two recent examples are AI algorithms to predict the survival of breast cancer (Stratipath Breast, Stratipath, Stockholm, Sweden) and colorectal cancer patients (Histotype Px Colorectal, DoMore Diagnostics, Oslo, Norway) directly from pathology slides. Based on publicly available information, these algorithms are presumably based on CNNs, not ViTs. Ultimately, these algorithms offer potential benefits in terms of efficiency and resource savings for diagnostic stakeholders, while at the same time offering the possibility of improved biomarkers for cancer patients. However, during this potential large-scale rollout of AI systems, it is important to ensure the robustness of these systems to artifacts and malicious interventions[43].

Here, we show that CNNs in computational pathology are susceptible to adversarial attacks far below the human perception threshold. We investigate two different and commonly used CNN models, ResNet50 (pretrained on Imagenet) and BiT[44], and show that both are equally susceptible to attacks. We show that existing mitigation strategies such as adversarial training and DBN do not provide universal mitigation. Addressing this issue, we explored the potential of ViTs to confer adversarial robustness to AI models. We show that ViTs perform on par with CNNs at baseline, and that they seem inherently more robust against adversarial attacks. In line with previous observations by Ma et al.[45], we also noticed that the bigger models with a higher number of trainable parameters are more vulnerable to adversarial attacks, but ViT is robust despite its large number of parameters. Although no AI models are universally and fully attack-proof, our study demonstrates that ViTs seem much more robust against common white-box and black-box attack types and that this is associated with a more robust behavior of the latent space compared to CNNs. Our findings add to a list of theoretical benefits of ViTs over CNNs and provide an argument to use ViTs as the core technology for AI products in computational pathology. The selection of end-to-end prediction pipelines in our study is motivated by the result of a recent benchmarking study which compared multiple state-of-the-art methods for computational pathology and showed that ResNet and ViT are outperforming many other common models in this field[23]. Also, our findings are in line with studies in non-medical domains which analyzed the robustness of ViTs in technical benchmark tasks[46,47].

A limitation of our study is the restriction to cancer use cases and classification tasks. A more difficult task such as predicting the response to therapy would have even more severe clinical implications and could not even be directly checked by a pathologist (as could the diagnostic classification tasks used in the study), since negative consequences for prognostic misclassifications have a time delay. Future work should also address other types of adversarial attacks, such as physical-world

attacks[17] or one-pixel attacks[48]. The uptake of newer AI models, such as text-image models, could also open vulnerabilities toward new types of adversarial attacks[49]. As multiple AI systems are nearing the diagnostic market, hardening these tools against established and emerging adversarial attacks should be a priority for the computational pathology research community in academia and industry[20].

## Methods

### Ethics statement

This study was performed in accordance with the Declaration of Helsinki. We performed a retrospective analysis of anonymized patient samples. In addition to publicly available data from "The Cancer Genome Atlas" (TCGA, https://portal.gdc.cancer.gov), we used a renal cell carcinoma dataset by the University of Aachen, Germany (ethics board of Aachen University Hospital, No. EK315/19) and a gastric cancer dataset by the University of Bern (ethics board at the University of Bern, Switzerland, no. 200/14). This study adheres to the MI-CLAIM[50] checklist (Suppl. Table 1). The need for informed consent was waived by the respective ethics commissions because this study was a retrospective anonymized analysis of archival samples and did not entail any contact with patients of any sort.

### Patient cohorts

We collected digital whole slide images (WSI) of H&E-stained tissue slides of renal cell carcinoma (RCC) from two patient cohorts: TCGA-RCC (*N* = 897 patients, Suppl. Fig. 1A), which was used as a training set and AACHEN-RCC (*N* = 249, Suppl. Fig. 1B), which was used as a test set. The objective was to predict RCC subtypes: clear cell (ccRCC), chromophobe (chRCC), and papillary (papRCC). In addition, we obtained H&E-stained slides of gastric cancer from two patient cohorts: TCGA-GASTRIC (*N* = 191 patients, Suppl. Fig. 1C) for training and BERN-GASTRIC (*N* = 249 patients, Suppl. Fig. 1D)[51] for testing. The objective was to predict the two major subtypes: intestinal and diffuse, according to the Laurén classification. Samples with mixed or indeterminate subtypes were excluded. Ground truth labels were obtained from the original pathology report.

### Image preprocessing

We tessellated the WSI into tiles (512 px edge length at 0.5 μm per pixel) which were color-normalized with the Macenko method[52]. No manual annotations were used. Background and blurry tiles were identified by having an average edge ratio smaller than 4, using the canny edge detection method, and were removed[53]. For each experiment, we selected 100 random tiles from each WSI. We used a classical weakly-supervised prediction workflow[38,54] in which each tile inherited the ground truth label from the WSI and tile-level predictions were averaged over the WSI at inference. Before each training run, the total number of tiles per class was equalized by random downsampling[2].

### Experimental design

First, we trained deep learning models on categorical prediction tasks in the training cohort and validated the performance in the test cohort. We used Deep Learning models, ResNet (specifically ResNet50, version 1), BiT (Big Transfer Model, also called ResNet50-v2)[55], a convolutional neural network (CNN), and Vision transformers (ViT)[56]. Then, we assessed the susceptibility of the trained models toward white- and black-box adversarial attacks. Finally, we evaluated mitigation strategies against adversarial attacks. One strategy was to attack the images in the training cohort, termed adversarially robust training. The other strategy, specific to CNNs, was to use dual batch normalization, as introduced recently by ref. 57.

### Implementation and analysis of adversarial attacks

For an image $X$ belonging to class $C_i$, an adversarial attack perturbs $X$ in such a way that the image is misclassified as $C_j, i \neq j$. We used six common types of attacks: (1) Fast Gradient Sign Method (FGSM)[58–60], a single-step gradient-based white-box attack; (2) Projected Gradient Descent (PGD)[61], a multi-step gradient-based white-box attack with attack strength $\epsilon$; (3) Fast Adaptive boundary (FAB)[62], a more generic type of gradient-based white-box attack; (4) Square attack[63], a black-box attack which places square-shaped updates at random positions on the input image; (5) AutoAttack (AA)[64], an ensemble of diverse parameter-free attacks (PGD, FAB, and Square); and (6) AdvDrop[65], which creates adversarial examples by dropping the high-frequency features from the image. To measure which amount of noise is detectable by humans, we randomly selected three tiles from the AACHEN-RCC dataset and attacked each of them with PGD with 50 different attack strengths (0 to 0.5). We presented these tiles to a blinded human observer (medical doctor) who subjectively classified the images as "no noise detectable" and "noise detectable". Subsequently, we determined the detection threshold by fitting a logistic regression model to the data. This analysis was run separately for noise generated with PGD on a ResNet and a ViT model. To visualize the adversarial noise, we subtracted the perturbed image from the original image, clipped at the 10th and 90th quantile for each color channel, and scaled between 0 and 255. In addition, we visualized the latent space of deep layer activations of CNNs and ViTs. The activation feature vectors of ResNet50 (1 × 2048) and ViT (1 × 768) were reduced to (1 × 2) by principal component analysis (PCA), and each component was scaled between 0 and 1. To quantify the separation between multiple classes in this latent space, we calculated the Euclidean distance[66] between all points of each class to the center of the corresponding classes and between the centers of classes. Additionally, we generated Gradient-weighted Class Activation Mapping (Grad-CAM) visualizations and investigated the role of adversarial attacks on the localization of important image regions by the models at baseline and after attacks.

### Statistics

The main statistical endpoint was the patient-wise micro-averaged area under the receiver operating curve (AUROC). 95% confidence intervals were obtained by 1000-fold bootstrapping based on sampling with replacement. The test dataset remained the same for the experiments between different models. All experiments were repeated five times with different random seeds. We reported the mean AUROC with standard deviation (SD) and median AUROC with interquartile range (IQR = $q_{75th} - q_{25th}$). Two-sided unpaired *t*-tests were used to compare sets of AUROCs between different deep learning models for the same experimental condition. No correction for multiple testing was applied. Furthermore, we calculated the attack success rate (ASR). The ASR quantified the effectiveness of an attack by calculating the degree of misclassification: if the model's prediction score for the perturbed image changes, the attack was deemed successful. The ASR was calculated for 450 randomly selected tiles per class from the AACHEN-RCC set.

### Reporting summary

Further information on research design is available in the Nature Research Reporting Summary linked to this article.

## Data availability

The data that support the findings of this study are mostly publicly available, in part proprietary datasets provided under collaboration agreements. All data (including histological images) from the TCGA database are available at https://portal.gdc.cancer.gov/. The cohort accession codes are TCGA-KIRC, TCGA-KIRP, TCGA-KICH, and TCGA-STAD. Access to the proprietary data can be requested from the respective study groups who independently manage data access for their study cohorts: Rupert Langer for BERN-GASTRIC, Roman D. Buelow and Peter Boor for AACHEN-RCC. The respective principal

investigators will respond within 4 weeks and will decide, according to the local institution's standards, if the data can be shared for research purposes under a dedicated collaboration agreement.

## Code availability

All source codes are publicly available: for image preprocessing[67], codes are available at https://github.com/KatherLab/preProcessing; for the baseline image analysis[23], codes are available at https://github.com/KatherLab/HIA, and for adversarial attacks, codes are available at https://github.com/KatherLab/Pathology_Adversarial[68]. Additional details are available in Supplementary Methods[69-74].

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

## Acknowledgements

J.N.K. is supported by the German Federal Ministry of Health (DEEP LIVER, ZMVI1-2520DAT111) and the Max-Eder-Program of the German Cancer Aid (grant #70113864). P.B. is supported by the DFG, German Research Foundation (Project-IDs 322900939, 454024652, 432698239, 445703531, and 445703531), European Research Council (ERC; Consolidator Grant AIM.imaging.CKD, No 101001791), Federal Ministry of Education and Research (STOP-FSGS-01GM1901A), and Federal Ministry of Economic Affairs and Energy (EMPAIA, No. 01MK2002A).

## Author contributions

N.G.L., D.T., and J.N.K. designed the study; N.G.L. and J.N.K. developed the software; N.G.L. performed the experiments; N.G.L., D.T., T.H., P.B., and J.N.K. analyzed the data; N.G.L. and M.v.T. performed statistical analyses; R.D.B., R.L., B.D., and P.B. provided clinical and histopathological data; all authors provided clinical expertise and contributed to the interpretation of the results. N.G.L., D.T., G.P.V., and J.N.K. wrote the manuscript, and all authors corrected the manuscript and collectively made the decision to submit it for publication.

## Funding

## Competing interests

J.N.K. declares consulting services for Owkin, France and Panakeia, UK. No other potential conflicts of interest are reported by any of the authors.

## Additional information

[1]Department of Medicine III, University Hospital RWTH Aachen, RWTH Aachen university, Aachen, Germany. [2]Department of Diagnostic and Interventional Radiology, University Hospital Aachen, Aachen, Germany. [3]Else Kroener Fresenius Center for Digital Health, Medical Faculty Carl Gustav Carus, Technical University Dresden, Dresden, Germany. [4]Department of Physics of Molecular Imaging Systems, Institute for Experimental Molecular Imaging, RWTH Aachen University, Aachen, Germany. [5]Institute of Pathology, University Hospital RWTH Aachen, Aachen, Germany. [6]Institute of Pathology, University of Bern, Bern, Switzerland. [7]Institute of Pathology and Molecular Pathology, Kepler University Hospital, Johannes Kepler University Linz, Linz, Austria. [8]Physics Institute III B, RWTH Aachen University, Aachen, Germany. [9]Fraunhofer Institute for Digital Medicine MEVIS, Aachen, Germany. [10]Hyperion Hybrid Imaging Systems GmbH, Aachen, Germany. [11]Medical Oncology, National Center for Tumor Diseases (NCT), University Hospital Heidelberg, Heidelberg, Germany. [12]Division of Pathology and Data Analytics, Leeds Institute of Medical Research at St James's, University of Leeds, Leeds, UK. [13]Department of Medicine 1, University Hospital and Faculty of Medicine Carl Gustav Carus, Technical University Dresden, Dresden, Germany. ✉e-mail: jakob_nikolas.kather@tu-dresden.de

