## [Peer Review File · Nature Communications]

Reviewers' Comments:

Reviewer #1:

Remarks to the Author:

Summary:

This work studies the adversarial vulnerability of commonly used deep learning models, convolutional neural networks (CNN) & vision transformers (ViT) in particular, for computational pathology. Two supervised categorical prediction tasks including RCC subtyping and gastric cancer subtyping were conducted with publicly available and privately collected datasets. Under standard training, the authors found that the CNN model is as good as the ViT model in terms of the AUROC on clean images, yet more vulnerable to 4 types of adversarial attacks (FGSM, PGD, FAB and Square). Under adversarial training, the ViT model also beats the CNN model at different confidence levels. The authors then draw the conclusion that ViT is more robust to adversarial attacks and that is because ViTs learn more robust latent representations than CNNs.

Strengths:

As AI systems are increasingly deployed to assist clinicians to make critical decisions, the adversarial vulnerability of deep learning-based models undoubtedly poses a serious threat to such systems which potentially could cause insurance fraud or even lethal consequences. This study makes an important step toward the fundamental understanding of the robustness/vulnerability discrepancy between different types of deep learning models. The experiments include both public and separately collected tissue slide images, verifying the transferability of the results in real-world applications. The analyses are also comprehensive, involving multiple attack methods, model types, prediction tasks, and training methods. The quartile and AUROC analyses also make the conclusion more convincing.

Weaknesses:

1. The main claim that ViTs are more robust than CNNs is not precise. This claim was made without taking the nature of different attacks into consideration. Out of the four tested attacks, FGSM, FAB and Square are much weaker than PGD, due to the way how they utilize the gradients. In the adversarial machine learning research community, adversarial robustness should be measured by the strongest attacks such as PGD and AutoAttack, and robustness against weak attacks is often deemed to be "not real" [1]. If measured by the strongest attack out of the four, PGD, then the ViT model is only slightly more robust than CNN, and is unknown against the commonly used robustness evaluation attack AutoAttack.
2. The unit of epsilon is not very clear, $\epsilon=0.20$ is with respect to $[0,1]$ or $[0,255]$? In Figure 2D, it appears that the images are in the range of $[0,255]$, as otherwise, $\epsilon=0.16$ should be a quite large (noticeable) perturbation.
3. The settings used in adversarial training and robustness evaluation are not consistent. I.e. the robust strength for training was $5e-3$, whereas it is much lower for evaluation, $0.25e-3$, $0.75e-3$, $1.5e-3$. Table 1 reports against $\epsilon=5e-3$, $1e-2$ and 0.2 , Figure 3A right reports $\epsilon < 1.4e-3$. It is quite difficult to evaluate the soundness of the conclusions drawn under different epsilons. For instance, Figure 3A right subfigure shows the ViT and DNB models are under (worse than) the ResNet model.
4. Several typos: type of deep neural network(s) (the last paragraph, page 3); ViTs are inherently robust (to) adversarial attacks (subsection title, page 9).
5. It would be interesting to see a discussion regarding the finding of Ma et al. [2]: large deep learning models are overparameterized for small medical datasets, and thus are more vulnerable on medical images than nature scene images. Is it the case with respect to different models (larger model is more vulnerable), i.e., CNN is more vulnerable because it has more parameters than ViT? Or how model complexity affects the conclusions.

[1] Croce, F., & Hein, M. (2020, November). Reliable evaluation of adversarial robustness with an ensemble of diverse parameter-free attacks. In International conference on machine learning (pp. 2206-2216). PMLR.

[2] Ma, X., Niu, Y., Gu, L., Wang, Y., Zhao, Y., Bailey, J., & Lu, F. (2021). Understanding adversarial attacks on deep learning based medical image analysis systems. *Pattern Recognition*, 110, 107332.

Reviewer #2:

Remarks to the Author:

This article examines adversarial attacks in the medical field. The authors investigate the vulnerability of convolutional neural networks (CNNs) to multiple types of white- and black-box attacks and demonstrate both attacks are capable of easily confounding CNNs in clinically relevant pathology tasks and diminishing classification performance. Classic adversarial robust training and dual batch normalization (DBN) are potential mitigation strategies for such attacks, but they require precise knowledge of the attack type used during inference. In the experiment section, the author show that vision transformers (ViTs) perform comparably to CNNs at baseline and are orders of magnitude more resistant to various white-box and black-box attacks. However, it does not fully address the core problem in preventing adversarial attacks: How the representation ability in neural network caused such difference. Moreover, the comparison method in the paper is too old-fashioned, since nowadays many state-of-the-art methods have been utilized into practice. More specific comments are as follows:

- 1) My main concern is the demonstration of the representation ability. Despite the experiments have exhibited the performance loss of the compared ViT and ResNet network, however, more insightful analysis should be given. For instance, the authors can utilize visualization techniques like Grad-CAM [1] and compare the results of each attack methods and dig into the causes.
- 2) The compared methods are old-fashioned. In recent years, many state-of-the-art methods have been developed like [2][3]. Also, the attack methods in the paper are somehow naive. Many effective such as network-based methods have been proposed during recent years. For instance, articles in [4][5] provide two potent methods for adversarial attacks. Moreover, the defense training methods also need to be updated. Existing study like [6] has provided a grounded method in defense training.
- 3) The paper claimed, "few paper have delved into the research of medical adversarial attack". However, similar literature studies are available like [7][8] and provide a relatively comprehensive review of the field.

[1] Selvaraju, Ramprasaath R., et al. "Grad-cam: Visual explanations from deep networks via gradient-based localization." *Proceedings of the IEEE international conference on computer vision*. 2017.

[2] Bohle, Moritz, Mario Fritz, and Bernt Schiele. "Convolutional Dynamic Alignment Networks for Interpretable Classifications." *Proceedings of the IEEE/CVF Conference on Computer Vision and Pattern Recognition*. 2021.

[3]Wu, Jianfang, et al. "Vision Transformer-based recognition of diabetic retinopathy grade." *Medical Physics* 48.12 (2021): 7850-7863.

[4] Duan, Ranjie, et al. "AdvDrop: Adversarial Attack to DNNs by Dropping Information." *Proceedings of the IEEE/CVF International Conference on Computer Vision*. 2021.

[5] Feng, Weiwei, et al. "Meta-Attack: Class-Agnostic and Model-Agnostic Physical Adversarial Attack." *Proceedings of the IEEE/CVF International Conference on Computer Vision*. 2021.

[6] Liao, Fangzhou, et al. "Defense against adversarial attacks using high-level representation guided denoiser." *Proceedings of the IEEE conference on computer vision and pattern recognition*. 2018.

[7] Ma, Xingjun, et al. "Understanding adversarial attacks on deep learning based medical image analysis systems." *Pattern Recognition* 110 (2021): 107332.

[8] Paul R, Schabath M, Gillies R, et al. Mitigating adversarial attacks on medical image understanding systems[C]//2020 IEEE 17th International Symposium on Biomedical Imaging (ISBI). IEEE, 2020: 1517-1521.

Reviewer #3:

Remarks to the Author:

This paper studies the adversarial robustness of deep learning models for computational pathology, and in particular, the comparisons between convolutional neural nets (ResNet50) and vision transformers (ViT B-16).

Evaluated on several white-box and black-box attacks (with a human doctor for discerning the presence of adversarial noise), the main findings are

1. ResNet is less robust than ViT
2. Adversarial training partially improves the robustness of ResNet, while being more beneficial for ViT
3. Case study on the visualization of adversarial perturbations and interpretations

The strengths (S's) and weaknesses (W's) are summarized below:

-S1: The studied task (computational pathology) is new in the adversarial machine learning community

-S2: State-of-the-art attack mitigation methods such as adversarial training are considered in the analysis

-S3: Paper is well-written and easy-to-follow

-W1: Some tested attacks are known to be very weak baselines (e.g., FGSM) and therefore the conclusion may be biased. Moreover, the considered attack is not state-of-the-art. The authors are suggested to evaluate against AutoAttack <<https://github.com/fra31/auto-attack>>, which is an ensemble of strong attacks, for a fair evaluation.

-W2: Taking ResNet (pre-trained on ImageNet) and ViT (pre-trained on a much larger dataset and fine-tuned on ImageNet) as two alternatives for comparison is not fair, because they are trained/pre-trained with image data at a very different scale. A fairer baseline should be Big Transfer model (BiT) v.s. ViT, as their pre-training data are similar. More critically, recent studies have shown that ViTs are not necessarily more robust than CNNs - see <https://arxiv.org/abs/2111.05464> for details. I suggest the readers follow the analysis and suggestions to make the comparison.

-W3: Given that both ResNet and ViT models were trained by the authors, it was unclear to me whether they are the best deep learning models for digital path pathology. To me, the paper presents a case study of transfer learning from an ImageNet pre-trained/fine-tuned model to the studied domain dataset, with some additional qualitative analysis that is tailored to this task. However, I am not so confident the conclusion is meaningful, because the studied models may not even be the right deep learning model for this task. Further validation and justification on why these two models (ResNet and ViT) are ideal candidates for this task are needed.

Summary:

Overall, this paper studies the adversarial robustness of two popular deep learning models on a new task and dataset. While I think there are some new insights, I also have several major concerns in terms of (1) lack of evaluation of state-of-the-art attacks (2) the two models were not treated fairly because the training/pre-training data are very different, and the conclusion may conflict with recent findings (3) why these two models are the right candidates for this task?

REVIEWER COMMENTS

Reviewer #1 (Remarks to the Author)

Comment: Summary: This work studies the adversarial vulnerability of commonly used deep learning models, convolutional neural networks (CNN) & vision transformers (ViT) in particular, for computational pathology. Two supervised categorical prediction tasks including RCC subtyping and gastric cancer subtyping were conducted with publicly available and privately collected datasets. Under standard training, the authors found that the CNN model is as good as the ViT model in terms of the AUROC on clean images, yet more vulnerable to 4 types of adversarial attacks (FGSM, PGD, FAB and Square). Under adversarial training, the ViT model also beats the CNN model at different confidence levels. The authors then draw the conclusion that ViT is more robust to adversarial attacks and that is because ViTs learn more robust latent representations than CNNs.

Response: Thank you very much for the concise summary. We have responded to all your points below.

Comment: Strengths: As AI systems are increasingly deployed to assist clinicians to make critical decisions, the adversarial vulnerability of deep learning-based models undoubtedly poses a serious threat to such systems which potentially could cause insurance fraud or even lethal consequences. This study makes an important step toward the fundamental understanding of the robustness/vulnerability discrepancy between different types of deep learning models. The experiments include both public and separately collected tissue slide images, verifying the transferability of the results in real-world applications. The analyses are also comprehensive, involving multiple attack methods, model types, prediction tasks, and training methods. The quartile and AUROC analyses also make the conclusion more convincing.

Response: Thank you very much for your very positive comments.

Weaknesses:

1. The main claim that ViTs are more robust than CNNs is not precise. This claim was made without taking the nature of different attacks into consideration. Out of the four tested attacks, FGSM, FAB and Square are much weaker than PGD, due to the way they utilize the gradients. In the adversarial machine learning research community, adversarial robustness should be measured by the strongest attacks such as PGD and AutoAttack, and robustness against weak attacks is often deemed to be “not real” (Croce and Hein 13--18 Jul 2020). If measured by the strongest attack out of the four, PGD, then the ViT model is only slightly more robust than CNN, and is unknown against the commonly used robustness evaluation attack AutoAttack.

Response: Thank you for suggesting that we should include additional stronger attacks. We have done this exactly as suggested, adding AutoAttack and AdvDrop to our experiments. In line with our previous findings, ViTs are much more robust than CNNs against these two attacks (revised Table 1, revised results section, lines 260-266).

2. The unit of epsilon is not very clear, $\epsilon=0.20$ is with respect to $[0,1]$ or $[0,255]$? In Figure 2D, it appears that the images are in the range of $[0,255]$, as otherwise, $\epsilon=0.16$ should be a quite large (noticeable) perturbation.

Response: In all our experiments, the pixel values in the images are in $[0,1]$, including Figure 2D. As stated by the reviewer, we show that any $\epsilon > 0.1$ is visible to the human eye (Suppl. Table 3). In all subsequent experiments, we use epsilon values well below this perception threshold. This is important because it shows that CNNs are highly susceptible even to sub-visual adversarial attacks.

3. The settings used in adversarial training and robustness evaluation are not consistent. I.e. the robust strength for training was $5e-3$, whereas it is much lower for evaluation, $0.25e-3$, $0.75e-3$, $1.5e-3$. Table 1 reports against $\epsilon=5e-3$, $1e-2$ and 0.2 , Figure 3A right reports $\epsilon < 1.4e-3$. It is quite difficult to evaluate the soundness of the conclusions drawn under different epsilons. For instance, Figure 3A right subfigure shows the ViT and DNB models are under (worse than) the ResNet model.

Response: We agree that the interpretation of our results in the initial manuscript was somewhat complicated by the use of multiple different epsilon values. We have streamlined this in the revised manuscript and always use $0.25e-3$, $0.75e-3$, and $1.50e-3$ as “low”, “medium”, and “high” intensity attacks, respectively. All our conclusions hold (revised Table 1) and ViT was more robust than CNNs in all experiments (revised results section, lines 260-266).

4. Several typos: type of deep neural network(s) (the last paragraph, page 3); ViTs are inherently robust (to) adversarial attacks (subsection title, page 9).

Response: This has been corrected, and we have carefully proof-read the revised manuscript.

5. It would be interesting to see a discussion regarding the finding of (Ma et al. 2021): large deep learning models are overparameterized for small medical datasets, and thus are more vulnerable on medical images than nature scene images. Is it the case with respect to different models (larger model is more vulnerable), i.e., CNN is more vulnerable because it has more parameters than ViT? Or how model complexity affects the conclusions.

Response: This is a very interesting comment which has prompted us to perform additional experiments. As suggested by the reviewer, we have repeated our experiments with an additional CNN, the big transfer model BiT (Kolesnikov et al. 2020). Compared to our baseline ResNet (pretrained on ImageNet (1k), 23 M parameters), BiT is pretrained on 21k images and has 68 M parameters. As a comparison, ViT was pretrained on a very big image data set and fine-tuned on ImageNet and has 85 M parameters. Although ViT has slightly more trainable parameters than BiT, ViT is much more robust towards adversarial

attacks (Table 1). We have added (Ma et al. 2021) as a reference (lines 319-321) and we have mentioned these new findings in line 266-270 in the results section.

Reviewer #2 (Remarks to the Author):

Comment: This article examines adversarial attacks in the medical field. The authors investigate the vulnerability of convolutional neural networks (CNNs) to multiple types of white- and black-box attacks and demonstrate both attacks are capable of easily confounding CNNs in clinically relevant pathology tasks and diminishing classification performance. Classic adversarial robust training and dual batch normalization (DBN) are potential mitigation strategies for such attacks, but they require precise knowledge of the attack type used during inference. In the experiment section, the authors show that vision transformers (ViTs) perform comparably to CNNs at baseline and are orders of magnitude more resistant to various white-box and black-box attacks. However, it does not fully address the core problem in preventing adversarial attacks: How the representation ability in neural networks caused such difference. Moreover, the comparison method in the paper is too old-fashioned, since nowadays many state-of-the-art methods have been utilized into practice. More specific comments are as follows.

Response: Thank you very much for the positive feedback on our manuscript. We have addressed all of your comments in detail, see below.

Comment: 1) My main concern is the demonstration of the representation ability. Despite the experiments have exhibited the performance loss of the compared ViT and ResNet network, however, more insightful analysis should be given. For instance, the authors can utilize visualization techniques like Grad-CAM (Selvaraju, Cogswell, and Das, n.d.) and compare the results of each attack methods and dig into the causes.

Response: To address the reviewer's question, we have now incorporated two approaches to investigate the neural network's representation ability. In addition to the visualization of the latent space (Figure 3), demonstrating a better class separation in ViT compared to CNN, we have performed Grad-CAM analyses, showing that the focus of ViTs on relevant image areas is more stable under attacks (new Suppl. Figure 4). We would like to thank the reviewer for suggesting these experiments, which further support the conclusions we make in our manuscript.

Comment: 2) The compared methods are old-fashioned. In recent years, many state-of-the-art methods have been developed like (Bohle, Fritz, and Schiele, n.d.) and (Wu et al. 2021)

Response: The selection of methods for weakly-supervised classification which we use in our paper is motivated by a recent comprehensive survey and benchmarking study (Ghaffari Laleh et al. 2022). Regarding the methods which were suggested by the reviewer:

- **(Bohle, Fritz, and Schiele):** This paper is presenting a very interesting model for interpretable classification of the images. The method uses a slightly

unconventional preprocessing (6-channel images). The method is not commonly used in applied studies investigating weakly-supervised prediction tasks in computational pathology.

- (Wu et al. 2021): This paper uses a ViT model which we also do in our study. The study investigates fundus images, not pathology images.

We have added this to the discussion section in lines 325-330.

Comment: Also, the attack methods in the paper are somehow naive. Many effective methods such as network-based methods have been proposed during recent years. For instance, articles in (Duan et al., n.d.) and (Feng et al., n.d.) provide two potent methods for adversarial attacks. Moreover, the defense training methods also need to be updated. Existing studies like (Liao et al., n.d.) have provided a grounded method in defense training.

Response: Thank you very much for your suggestion to include additional state-of-the-art attack methods in our study. We have done exactly as suggested by the reviewer and have added AdvDrop and AutoAttack (revised Table 1). MetaAttack could not be included because it is specific to graph-based methods. These additional experimental results provide further support for our conclusion, as we have mentioned in the revised results section (lines 262-266) Regarding state-of-the-art defense methods, we use a recent high-profile publication (Han et al. 2021). Liao et al. propose another defense method which is specific to CNNs. In our study, we show that ViTs are robust towards adversarial attacks even without specific defense mechanisms, which is the key insight presented by our study.

Reviewer #3 (Remarks to the Author):

Comment: This paper studies the adversarial robustness of deep learning models for computational pathology, and in particular, the comparisons between convolutional neural nets (ResNet50) and vision transformers (ViT B-16). Evaluated on several white-box and black-box attacks (with a human doctor for discerning the presence of adversarial noise), the main findings are

1. ResNet is less robust than ViT
2. Adversarial training partially improves the robustness of ResNet, while being more beneficial for ViT
3. Case study on the visualization of adversarial perturbations and interpretations

The strengths (S's) and weaknesses (W's) are summarized below:

- S1: The studied task (computational pathology) is new in the adversarial machine learning community
- S2: State-of-the-art attack mitigation methods such as adversarial training are considered in the analysis
- S3: Paper is well-written and easy-to-follow

Response: Thank you very much for this positive evaluation of our manuscript.

Comment: -W1: Some tested attacks are known to be very weak baselines (e.g., FGSM) and therefore the conclusion may be biased. Moreover, the considered attack is not state-of-the-art. The authors are suggested to evaluate against AutoAttack <https://github.com/fra31/auto-attack>, which is an ensemble of strong attacks, for a fair evaluation.

Response: **As suggested by the reviewer, we have repeated our experiments with AutoAttack (revised Table 1). The new data provide additional support for our conclusions.**

Comment: W2: Taking ResNet (pre-trained on ImageNet) and ViT (pre-trained on a much larger dataset and fine-tuned on ImageNet) as two alternatives for comparison is not fair, because they are trained/pre-trained with image data at a very different scale. A fairer baseline should be Big Transfer model (BiT) v.s. ViT, as their pre-training data are similar. More critically, recent studies have shown that ViTs are not necessarily more robust than CNNs - see <https://arxiv.org/abs/2111.05464> for details. I suggest the readers follow the analysis and suggestions to make the comparison.

Response: **This is a very valid point. We have addressed this by repeating our manuscript with BiT as suggested by the reviewer (revised Table 1). These data show that BiT is even more susceptible to adversarial attacks, further strengthening our conclusions.**

Comment: -W3: Given that both ResNet and ViT models were trained by the authors, it was unclear to me whether they are the best deep learning models for digital path pathology. To me, the paper presents a case study of transfer learning from an ImageNet pre-trained/fine-tuned model to the studied domain dataset, with some additional qualitative analysis that is tailored to this task. However, I am not so confident the conclusion is meaningful, because the studied models may not even be the right deep learning model for this task. Further validation and justification on why these two models (ResNet and ViT) are ideal candidates for this task are needed.

Response: **Thank you very much for bringing this up. We are very confident that the selected models are the best deep learning models for this task. This is based on a systematic survey of supervised classification pipelines in computational pathology, as well as a systematic benchmark study which we recently published in “Medical Image Analysis”. (Ghaffari Laleh et al. 2022) The choice of ResNet as the SOTA for classification was also demonstrated in other studies by our team (Kather et al. 2019, 2020) and by other groups (Srinidhi, Ciga, and Martel 2021). On the other hand, we are aware of the increasing popularity of the ViTs in computational pathology (Ghaffari Laleh et al. 2022; Chen et al. 2022). This motivated us to compare the robustness of the ViTs to the CNN (mainly ResNet, because of its popularity). We mentioned these points in the discussion section, lines 325-330.**

Comment: Summary: Overall, this paper studies the adversarial robustness of two popular deep learning models on a new task and dataset. While I think there are some new insights, I also have

several major concerns in terms of (1) lack of evaluation of state-of-the-art attacks (2) the two models were not treated fairly because the training/pre-training data are very different, and the conclusion may conflict with recent findings (3) why these two models are the right candidates for this task?

Response: We would like to thank you and mention that your useful comments helped us to improve the quality of our manuscript. As we described in the above sections, we added more SOTA attacks to our attack pool and compared their performance. We added BiT which was trained on a bigger data set and therefore provided a more unbiased comparison of the performances. Finally, we extensively discussed the reasons behind the selection of residual models and ViTs.

References for “Response to Reviewers” letter only

- Bohle, Fritz, and Schiele. n.d. “Convolutional Dynamic Alignment Networks for Interpretable Classifications.” *Proceedings of the IEEE/CVF*.
http://openaccess.thecvf.com/content/CVPR2021/html/Bohle_Convolutional_Dynamic_Alignment_Networks_for_Interpretable_Classifications_CVPR_2021_paper.html.
- Chen, Richard J., Chengkuan Chen, Yicong Li, Tiffany Y. Chen, Andrew D. Trister, Rahul G. Krishnan, and Faisal Mahmood. 2022. “Scaling Vision Transformers to Gigapixel Images via Hierarchical Self-Supervised Learning.” *arXiv [cs.CV]*. arXiv.
<http://arxiv.org/abs/2206.02647>.
- Croce, Francesco, and Matthias Hein. 13–18 Jul 2020. “Reliable Evaluation of Adversarial Robustness with an Ensemble of Diverse Parameter-Free Attacks.” In *Proceedings of the 37th International Conference on Machine Learning*, edited by Hal Daumé Iii and Aarti Singh, 119:2206–16. Proceedings of Machine Learning Research. PMLR.
- Duan, Chen, Niu, and Yang. n.d. “Advdrop: Adversarial Attack to Dnns by Dropping Information.” *Proceedings of the Estonian Academy of Sciences. Biology, Ecology = Eesti Teaduste Akadeemia Toimetised. Bioloogia, Okoloogia*.
http://openaccess.thecvf.com/content/ICCV2021/html/Duan_AdvDrop_Adversarial_Attack_to_DNNs_by_Dropping_Information_ICCV_2021_paper.html.
- Feng, Wu, Zhang, and Zhang. n.d. “Meta-Attack: Class-Agnostic and Model-Agnostic Physical Adversarial Attack.” *Proceedings of the Estonian Academy of Sciences. Biology, Ecology = Eesti Teaduste Akadeemia Toimetised. Bioloogia, Okoloogia*.
http://openaccess.thecvf.com/content/ICCV2021/html/Feng_Meta-Attack_Class-Agnostic_and_Model-Agnostic_Physical_Adversarial_Attack_ICCV_2021_paper.html.
- Ghaffari Laleh, Narmin, Hannah Sophie Muti, Chiara Maria Lavinia Loeffler, Amelie Echle, Oliver Lester Saldanha, Faisal Mahmood, Ming Y. Lu, et al. 2022. “Benchmarking Weakly-Supervised Deep Learning Pipelines for Whole Slide Classification in Computational Pathology.” *Medical Image Analysis* 79 (July): 102474.
- Han, Tianyu, Sven Nebelung, Federico Pedersoli, Markus Zimmermann, Maximilian Schulze-Hagen, Michael Ho, Christoph Haarbuerger, et al. 2021. “Advancing Diagnostic Performance and Clinical Usability of Neural Networks via Adversarial Training and Dual Batch Normalization.” *Nature Communications* 12 (1): 4315.
- Kather, Jakob Nikolas, Lara R. Heij, Heike I. Grabsch, Chiara Loeffler, Amelie Echle, Hannah Sophie Muti, Jeremias Krause, et al. 2020. “Pan-Cancer Image-Based Detection of Clinically Actionable Genetic Alterations.” *Nature Cancer* 1 (8): 789–99.
- Kather, Jakob Nikolas, Alexander T. Pearson, Niels Halama, Dirk Jäger, Jeremias Krause, Sven H. Loosen, Alexander Marx, et al. 2019. “Deep Learning Can Predict Microsatellite Instability Directly from Histology in Gastrointestinal Cancer.” *Nature Medicine* 25 (7): 1054–56.
- Kolesnikov, Alexander, Lucas Beyer, Xiaohua Zhai, Joan Puigcerver, Jessica Yung, Sylvain Gelly, and Neil Houlsby. 2020. “Big Transfer (BiT): General Visual Representation Learning.” In *Computer Vision – ECCV 2020*, 491–507. Springer International Publishing.
- Liao, Liang, Dong, and Pang. n.d. “Defense against Adversarial Attacks Using High-Level Representation Guided Denoiser.” *Proceedings of the Estonian Academy of Sciences. Biology, Ecology = Eesti Teaduste Akadeemia Toimetised. Bioloogia, Okoloogia*.
http://openaccess.thecvf.com/content_cvpr_2018/html/Liao_Defense_Against_Adversarial_CVPR_2018_paper.html.
- Ma, Xingjun, Yuhao Niu, Lin Gu, Yisen Wang, Yitian Zhao, James Bailey, and Feng Lu. 2021. “Understanding Adversarial Attacks on Deep Learning Based Medical Image Analysis Systems.” *Pattern Recognition* 110 (February): 107332.
- Selvaraju, Cogswell, and Das. n.d. “Grad-Cam: Visual Explanations from Deep Networks via

Gradient-Based Localization.” *Proceedings of the Estonian Academy of Sciences. Biology, Ecology = Eesti Teaduste Akadeemia Toimetised. Bioloogia, Okoloogia*.
http://openaccess.thecvf.com/content_iccv_2017/html/Selvaraju_Grad-CAM_Visual_Explanations_ICCV_2017_paper.html.

Srinidhi, Chetan L., Ozan Ciga, and Anne L. Martel. 2021. “Deep Neural Network Models for Computational Histopathology: A Survey.” *Medical Image Analysis* 67 (January): 101813.

Wu, Jianfang, Ruo Hu, Zhenghong Xiao, Jiaxu Chen, and Jingwei Liu. 2021. “Vision Transformer-Based Recognition of Diabetic Retinopathy Grade.” *Medical Physics* 48 (12): 7850–63.

Reviewers' Comments:

Reviewer #1:

Remarks to the Author:

Thanks for the detailed response and revision. Most of my concerns have been properly addressed. But there are still several issues that need to be fixed (see below).

1. Figure 2C should be updated to include the patterns generated by AutoAttack (may be the first two AutoPGD attacks) and AdvDrop.

2. All figures need to be updated to include the newly added attacks and models: AutoAttack, BiT, etc.

3. The authors clarified that we show that any $\epsilon > 0.1$ is visible to the human eye (Suppl. Table 3). But why $1.50e-3$ was used for adversarial training rather than 0.1 ? $1.50e-3$ is too small to support the claims. The authors should also verify their results under larger epsilon close to 0.1 , as the attacker has no reason to restrict the attack strength to $1.50e-3$.

Reviewer #2:

Remarks to the Author:

I appreciate the authors' efforts addressing my comments. Most of minor comments have been properly addressed and revised accordingly. However, the major concerns of the insightful analysis are still not profound and this paper does not address the core problem of preventing adversarial attacks. Although it's good to see such as analysis specifically for pathology image tasks, the essential difference from other natural image tasks is not significant.

Reviewer #3:

Remarks to the Author:

The authors have fully addressed my concerns about the previous version.

REVIEWER COMMENTS

Reviewer #1 (Remarks to the Author):

Thanks for the detailed response and revision. Most of my concerns have been properly addressed. But there are still several issues that need to be fixed (see below).

Response: Thank you very much for the positive assessment. We have carefully addressed all of the remaining points, see below.

1. Figure 2C should be updated to include the patterns generated by AutoAttack (may be the first two AutoPGD attacks) and AdvDrop.

Response: As requested, we have updated Figure 2C and added the noise pattern from all the attacks which we used in our experiments.

2. All figures need to be updated to include the newly added attacks and models: AutoAttack, BiT, etc.

Response: Done as requested:

- **Figure 1: not applicable (no attacks)**
- **Figure 2: We added all new attacks (panel C)**
- **Figure 3: To avoid overcrowding this figure, we added all new attacks to a dedicated supplementary figure (new Suppl. Figure 3). The results provide further evidence supporting our conclusions.**

3. The authors clarified that we show that any $\epsilon > 0.1$ is visible to the human eye (Suppl. Table 3). But why $1.50e-3$ was used for adversarial training rather than 0.1 ? $1.50e-3$ is too small to support the claims. The authors should also verify their results under larger ϵ close to 0.1 , as the attacker has no reason to restrict the attack strength to $1.50e-3$.

Response: Our choices of ϵ was the same as in Han et al., Nature Communications, 2021. Yet, the reviewer's request prompted us to perform additional experiments with an ϵ of 0.1 (see revised Table 1). At this high ϵ , most models are severely impaired. Because 0.1 is the threshold for human perception of adversarial attacks, these attacks are of a low practical relevance. In contrast, attacks in the low sub-visual range (e.g. $\epsilon = 1.5e-3$) are very hard to detect and still detrimental to the model performance. This is what we have to make our models robust against, and we show that Vision Transformers are inherently much more robust (with any common type of adversarial attack). Hence, our findings are a huge improvement to Han et al., Nature Communications, 2021, and will set the stage for routinely incorporating adversarial robustness into computational pathology models.

Reviewer #2 (Remarks to the Author):

I appreciate the authors' efforts addressing my comments. Most of the minor comments have been properly addressed and revised accordingly. However, the major concerns of the insightful analysis are still not profound and this paper does not address the core problem of preventing adversarial attacks.

Response: Thank you very much for the overall positive evaluation of our work. Our study shows that using vision transformers instead of CNNs is a pragmatic way to limit the danger of adversarial attacks. At the same time, our work paves the way for future work to develop new specific solutions to prevent adversarial attacks in computational pathology.

Although it's good to see such an analysis specifically for pathology image tasks, the essential difference from other natural image tasks is not significant.

Response: Thank you very much. Computational pathology is exponentially growing, both in terms of academic publications and clinically approved (CE/IVD or FDA approved) algorithms. We are convinced that our domain-specific insights are highly relevant to researchers in academia and industry in this field.

Reviewer #3 (Remarks to the Author):

The authors have fully addressed my concerns about the previous version.

Response: Thank you very much for the positive evaluation of our work. No action needed.